# Mining the Roles of Cucumber DUF966 Genes in Fruit Development and Stress Response

**DOI:** 10.3390/plants11192497

**Published:** 2022-09-23

**Authors:** Jie Tian, Yiting Li, Yifeng Hu, Qiwen Zhong, Junliang Yin, Yongxing Zhu

**Affiliations:** 1Qinghai Key Laboratory of Vegetable Genetics and Physiology, Academy of Agriculture and Forestry Sciences of Qinghai University, Xining 810016, China; 2Engineering Research Center of Ecology and Agricultural Use of Wetland, Ministry of Education, Yangtze University, Jingzhou 434025, China

**Keywords:** nucleus localization, evolutionary analysis, expression profiling

## Abstract

DUF966 genes are widely found in monocotyledons, dicotyledons, mosses, and other species. Current evidence strongly suggests that they are involved in growth regulation and stress tolerance in crops. However, their functions in cucumbers remain unexplored. Here, cucumber CsDUF966 was systemically identified and characterized using bioinformatics. Eight CsDUF966 genes were identified in the cucumber genome. These were phylogenetically separated into three groups. All CsDUF966 proteins were hydrophilic and localized to the nucleus. Moreover, three acidic and five basic proteins were identified. Evolutionary analysis of DUF966 between cucumber and 33 other Cucurbitaceae species/cultivars revealed that most CsDUF966 genes were conserved, whereas CsDUF966_4.c and CsDUF966_7.c were positively selected among the five cucumber cultivars. Expression profiling analysis showed that CsDUF966 had variable expression patterns, and that miRNA164, miRNA166, and Csa-novel-35 were involved in the post-transcriptional regulation of CsDUF966_4.c and CsDUF966_7.c. The expression of CsDUF966_4.c and CsDUF966_7.c, which were under strong neofunctionalization selection, was strictly regulated in fruit and tissues, including seeds, pericarps, peels, and spines, suggesting that these genes are fruit growth regulators and were strongly selected during the cucumber breeding program. In conclusion, the results reveal the roles of CsDUF966s in regulating cucumber fruit development and lay the foundation for further functional studies.

## 1. Introduction

The domain of unknown function (DUF, PF06136) is a term designated for many domains in the Pfam database that have not been experimentally validated [1]. These domains exhibit two distinct features: (1) relatively conserved amino acid residues comprise the domains and (2) the biological function of the domain is currently unknown [2]. In recent years, systemic function analysis has been performed on DUF superfamily members, and the biological roles of some have been deciphered. Genes containing DUF domains participate in plant growth and developmental processes. For example, *OsREL2*, the first reported DUF domain-containing protein in rice, can control leaf rolling [3]. *At2g23470*, a member of the DUF647 family in *Arabidopsis*, plays a role in controlling fertility [4]. *PdDUF231A*, a member of the DUF231 family, was found to be involved in xylan acetylation and cellulose biosynthesis in *Populus deltoides* [5]. Furthermore, genes containing DUF domains have been reported to be involved in stress tolerance. *TaWTF1*, a wheat gene encoding the DUF860 protein, is modulated by heat stress at the seedling and flowering stages, with hormone- and stress-responsive elements being identified in its promoter region [6]. Rice OsDUF936.3, OsDUF936.5, and OsDUF936.6 were significantly upregulated by salt stress, and the overexpression of OsDUF936.6 in *Escherichia coli* greatly improved its tolerance to salt stress [7].

DUF966, a member of the DUF superfamily, is widely found in monocotyledons, dicotyledons, mosses, and other species [2]. DUF966 family members typically contain one or two highly conserved DUF966 domains. Previous studies have indicated that they are involved in abiotic stress tolerance. For example, overexpression of *OsDSR2* (*DUF966-stress repressive gene 2*) in rice can dramatically increase the sensitivity of plants to salt and drought stress and reduce susceptibility to abscisic acid (ABA) [8]. In *Arabidopsis*, the DUF966 gene *AtST39* (*At3g46110*) was found to strengthen salt and drought tolerance by positively regulating resistance pathways [8].

Cucumber is one of the most important vegetable crops cultivated worldwide [9]. However, the quality and yield of cucumbers are easily affected by adverse environmental stresses [10], such as drought, salinity, and nutrient deficiency, since their roots are largely distributed in shallow soil [11]. Breeding is an effective way to improve crop yield and resistance to environmental stresses [12,13]. The molecular genetics approaches, like mutation, marker assisted selection (MAS), genomics, recombinant DNA technology, targeted induced local lesions in the genome (TILLING), and virus induced gene silencing (VIGS), were adapted by breeders to develop effective resistance in crop plants in a shorter time [14]. In the past few years, the complete genome sequencing of cucumber and the discovery of related genetic information have provided the opportunity to make progress for breeding tolerance to the specific environmental stress [15], andidentify protein families that function in response to various stresses at the genome-wide level [16]. Although DUF966 genes play essential roles in crops [2], to the best of our knowledge, their functions remain unexplored in cucumber. In this study, CsDUF966 (*Cucumis sativus* DUF966) family members were systemically identified and characterized using bioinformatic methods. Chromosome distribution, gene structure, motifs, *cis*-acting elements, expression patterns, and subcellular localization were analyzed. Our results (1) provide valuable information for understanding the classification and characterization of DUF966 genes, (2) reveal the roles of CsDUF966s in regulating cucumber fruit development and stress response, and (3) lay the foundation for further functional studies of CsDUF966s and their application in cucumber breeding programs.

## 2. Results

### 2.1. Identification of CsDUF966 Genes from Cucumber

Eight CsDUF966 genes were identified in the cucumber reference genome (Appendix A). They were named according to the following rules: (1) CsDUF966 stands for the *Cucumis sativus* L. DUF966 gene family; (2) the ‘_number’ represents the gene number; (3) ‘.a/b/c’ represents the group to which the gene belongs (Table 1). To better understand the phylogenetic relationships of DUF966 proteins between cucumber and other plants, a phylogenetic tree was constructed using DUF966 sequences from cucumber, rice, pepper, and *Arabidopsis* (Figure 1). These DUF966 proteins were separated into three groups (a, b, and c). CsDUF966 proteins were present in all three groups. Groups a, b, and c contained one (CsDUF966_6.a), two (CsDUF966_1.b and CsDUF966_5.b), and five members, respectively (Figure 1). Amino acid length ranged from 199 to 520, with an average of 336 (Table 1). All CsDUF966 proteins were unstable, with an instability index > 40. The isoelectric point varied from 5.31 to 8.78, with three being acidic and five being basic. The grand average of hydropathicity values for all CsDUF966 proteins was less than 0, suggesting that they are all hydrophilic proteins. A subcellular localization prediction showed that all eight CsDUF966 proteins were localized in the nucleus.

### 2.2. Chromosomal Distribution and Ka/Ks Analysis of CsDUF966

The eight identified CsDUF966 genes were distributed on five chromosomes and two scaffolds, each containing one or two CsDUF966 genes (Figure 2A). Only one tandem duplication event was detected between the gene pair of CsDUF966_Un1.c and CsDUF966_Un2.c. The Ka/Ks values of DUF966 homolog pairs between cucumber and 33 other Cucurbitaceae species/cultivars were calculated (Figure 2B). The results showed that DUF966 genes were conserved among cucumber cv. Chinese long and other Cucurbitaceae species, as the Ka/Ks values between the majority of homolog pairs were less than 0.5. Nevertheless, some DUF966 genes were positively selected among cv. Chinese long and four other cucumber cultivars (Figure 2B, Appendix A), since Ka/Ks values, such as in homologous pairs of CsDUF966_4.c and Cucsat.G12087.T2, CsDUF966_7.c and Cucsat.G3032.T6, CsDUF966_4.c and CsGy4G004530.2, CsDUF966_7.c and CsGy7G018050.2, were larger than 1, suggesting that CsDUF966_4.c and CsDUF966_7.c are under strong neofunctionalization selection among different cucumber cultivars (Figure 2B).

### 2.3. Gene Structure and Motif Composition of CsDUF966s

Gene structure analysis showed that all CsDUF966 genes contained an average of 3.6 introns (range: two to seven) and 4.5 exons (range: three to seven) (Figure 3B). Motif analysis showed six to twelve motifs in each of the CsDUF966 proteins (Figure 3C). Proteins clustered close to branches in the phylogenetic tree had similar motif numbers and structures. Among them, motifs one, two, seven, and eight comprised the conserved DUF966 domain, and other motifs matched no known functional domains.

A three-dimensional (3D) model prediction analysis revealed that the CsDUF966 proteins showed three main types of crystal structure (Figure 3D). As shown in Figure 3D, compared with the other seven proteins, CsDUF966_4.c exhibited a distinct 3D structure. The remaining seven proteins exhibited similar structures. However, CsDUF966_Un1.c, CsDUF966_Un2.c, and CsDUF966_Un3.c had the same 3D structure, while CsDUF966_1. b, CsDUF966_5.b, CsDUF966_6.a, and CsDUF966_7.c showed the same 3D structure. A tiny 3D structural difference was observed in the C-terminal of these two groups of proteins. Further analysis showed that CsDUF966 proteins were composed mainly of *α*-helix, *β*-turn, and random coils, though the proportions of these three types of structures were different (Figure 3D). The proportion of random coils was the largest (43.01–62.31%), followed by the *α*-helix (16.08–30.47%), and the smallest share was the *β*-turn (5.58–14.7%).

### 2.4. Cis-Acting Element Analysis of CsDUF966s

In total, 40 kinds of *cis*-acting elements were detected in the upstream 1500 bp region of eight CsDUF966 genes, and the numbers of different *cis*-acting elements were marked in the box. These elements can be divided into three groups. Specifically, 25 *cis*-acting elements belong to “abiotic/biotic stress”, 11 belong to “phytohormone response”, and four belong to “growth and development”, respectively (Figure 4, Appendix A). Among them, a large number of TATA-box and CAAT-box were commonly detected in each CsDUF966 promoter, indicating that CsDUF966 genes widely participate in the growth and development regulation of cucumber. In addition, phytohormone responsive elements, such as the abscisic acid (ABRE), jasmonic acid (TGACG-motif, CGTCA-motif), gibberellins (P-box, GARE-motif), salicylic acid (TCA-element), and auxin (TGA-element), were found in CsDUF966 promoters, suggesting the potential ability of CsDUF966 genes to respond to phytohormones. Furthermore, abiotic/biotic stress responsive elements, such as light (3-AF1 binding site, AAAC-motif, ACE, AE-box, AT1-motif, Box 4, chs-CMA1a, GATA-motif, G-box, GT1-motif, GTGGC-motif, I-box, MRE, TCT-motif), anaerobic induction (ARE, GC-motif), low-temperature (LTR), defense and stress (TC-rich repeats), elicitor-mediated activation (AT-rich sequence), were detected, implying the roles of CsDUF966 genes in regulating the response of cucumber to stress conditions (Figure 4).

### 2.5. Transcriptomic Profiling of CsDUF966 Genes

As shown in Figure 5, an expression profiling analysis showed that CsDUF966 genes exhibited variable expression patterns (Appendix A). CsDUF966_1.b, CsDUF966_5.b, and CsDUF966_6.a were widely expressed in most cultivars, tissues, and treatments, whereas CsDUF966_4.c, CsDUF966_7.c, CsDUF966_Un1.c, CsDUF966_Un2.c, and CsDUF966_Un3.c were highly expressed in fruits and tissues including seeds, pericarps, peels, and spines; expressed at low to high levels in fruits, roots, and shoots; and not expressed in leaves. In shoots, several genes, such as CsDUF966_Un1.c, CsDUF966_Un2.c, CsDUF966_Un3.c, and CsDUF966_4.c, were responsive to low and high temperatures, short and long photo periods, and gibberellic acid. In leaves, CsDUF966_1.b, CsDUF966_5.b, and CsDUF966_6.a were responsive to powdery mildew. Furthermore, CsDUF966_1.b and CsDUF966_5.b showed different expression patterns in root differentiation, elongation, and meristematic zones.

### 2.6. Post-Transcription Regulation of miRNA to CsDUF966

As shown in Figure 6, ten cucumber miRNAs were predicted to target the five CsDUF966 genes (CsDUF966_1.b, CsDUF966_4.c, CsDUF966_6.a, CsDUF966_7.c, and CsDUF966_Un3.c) (Appendix A). All miRNAs regulate the expression of CsDUF966 genes through the cleavage effect. Notably, CsDUF966_4.c was potentially targeted by eight miRNAs, including csa-miR-166, csa-miR-166k, csa-miR-166t-1, csa-miR-166t-2, csa-miR-166v, csa-miR-166u, csa-miR-166x, and csa-miR-164c. CsDUF966_Un3.c is targeted by csa-miR-164c, and CsDUF966_6.a is targeted by csa-miR-827c. Additionally, csa-novel-35 was predicted to target CsDUF966_1.b and CsDUF966_7.c simultaneously.

### 2.7. Experimental Validation the Subcellular Localization of CsDUF966s

In silico prediction indicated that all CsDUF966 proteins were localized in the nucleus. To verify this result and reveal CsDUF966s function patterns, eight proteins were fused to GFP, and a subcellular localization analysis was performed. As shown in Figure 7, consistent with the prediction, all CsDUF966s were localized in the nucleus, revealing that CsDUF966s contribute to their potential biological roles in the nucleus.

### 2.8. Expression Analysis of CsDUF966

In order to explore the response of CsDUF966s to different stresses, the expression of CsDUF966s genes (CsDUF9661.b, CsDUF9664.c, CsDUF966_5.b, CsDUF966_6.a, CsDUF966_7.c, and CsDUF966_Un2.c) under heat, chilling, salt stress, and drought stresses were detected by qRT-PCR.

As shown in the Figure 8, in the root, the expression levels of CsDUF966_5.a, CsDUF966_7.c, and CsDUF966_Un2.c were greatly up-regulated under heat stress, but were decreased by chilling, cold, and PEG treatments. The expression level of CsDUF966_1.b was decreased by all treatments. PEG treatments increased the expression levels of CsDUF966_4.c and CsDUF966_5.b. In the stem, the expression levels of CsDUF966_1.b, CsDUF966_4.c, CsDUF966_5.b, CsDUF966_6.a, and CsDUF966_7.c were largely decreased by cold stress. Heat stress decreased the expression levels of CsDUF966_1.b and CsDUF966_6.a, but increased the expression levels of CsDUF966_5.b and CsDUF966_Un2.c. PEG treatments increased the expression levels of CsDUF966_1.b, CsDUF966_4.c, and CsDUF966_6.a. NaCl stress increased the expression level of CsDUF966_1.b, but decreased the expression levels of CsDUF966_6.a and CsDUF966_Un2.c. In the leaf, the expression levels of CsDUF966_1.b and CsDUF966_2.c were decreased by all treatments. The expression levels of CsDUF966_5.b and CsDUF966_6.a were decreased by cold and NaCl treatments. Heat stress increased the expression levels of CsDUF966_4.c and CsDUF966_5.b, and greatly decreased those of CsDUF966_1.b and CsDUF966_2.c.

## 3. Discussion

DUF966 is widely found in monocotyledons, dicotyledons, mosses, and other species [2]. Although the biological function of DUF966 family members is not fully understood, available evidence strongly suggests that they are involved in growth regulation and stress tolerance in plants. Despite this, to the best of our knowledge, their functions in cucumbers, one of the most important vegetable crops cultivated worldwide, remain unexplored [9].

In this study, CsDUF966 (*Cucumis sativus* DUF966) family members were systemically identified and characterized using bioinformatic methods. In total, eight CsDUF966 genes were identified from the cucumber reference genome Chinese long v3 [17]. These were phylogenetically separated into three groups (a, b, and c). Group a contains one member (CsDUF966_6.a), group b contains two members (CsDUF966_1.b and CsDUF966_5.b), and group c contains the remaining five members (Figure 1). Consistent with their transcription factor functions, all of these proteins were predicted to be localized in the nucleus. To further experimentally validate the nucleus localization, eight CsDUF966 proteins were labeled with GFP and observed by fluorescence microscopy. These results clearly indicate that CsDUF966s are localized in the nucleus (Figure 7).

Interestingly, eight CsDUF966 genes were highly conserved between cucumber and 29 other Cucurbitaceae species/cultivars, whereas certain genes, such as CsDUF966_4.c and CsDUF966_7.c, were under strong positive selection among the five cucumber cultivars (Ka/Ks ratio larger than 1) (Figure 2B). It should also be noted that, consistent with positive selection, CsDUF966_4.c has a distinct 3D structure when compared with the other seven CsDUF966, implying potential neofunctionalization selection during the breeding process of cucumber. Considering that the expression pattern of CsDUF966_4.c was strongly upregulated in fruit and tissues including seeds, pericarps, peels, and spines; was not expressed in leaves; and was responsive to low and high temperature, short and long photo period, and gibberellic acid in shoots, it is speculated that CsDUF966_4.c plays roles in fruit growth and development and stress response. This further explains CsDUF966_4.c as a key selection trait during cucumber breeding.

Based on 3D structural analysis, CsDUF966_7.c, CsDUF966_1.b, CsDUF966_5.b, and CsDUF966_6.a exhibited the same protein structure, which implies that they could have similar gene functions. However, unlike for CsDUF966_1.b, CsDUF966_5.b, and CsDUF966_6.a, a Ka/Ks analysis indicated that CsDUF966_7.c was under strong neofunctionalization selection among different cucumber cultivars. From the perspective of evolution, it seems that during cucumber cultivar breeding and selection, CsDUF966_1.b, CsDUF966_5.b, and CsDUF966_6.a were under negative selection to maintain their function, whereas CsDUF966_7.c was under positive selection to fulfill neofunctionalization. Furthermore, as shown in Figure 5A, the expression of CsDUF966_7.c was precisely regulated in fruits. For example, it was highly expressed in peels 8 d post-pollination in cv. Gy14 and Vlaspik, but not in peels at 16 d. It was highly expressed in the pericarp tissue of cv. NCG121 and cv. NCG122 pericarp tissue on the 6th day before flowering expressed at a low level in ovary tissue on the day of flowering, highly expressed in 2-4 cm early fruits of the long fruit near-isogenic line 408, and expressed at a low level in the short fruit line 409 (Figure 5), suggesting that CsDUF966_7.c may participate in the regulation of cucumber fruit development. Integrating fruit expression patterns and positive selection pressure, it is speculated that CsDUF966_7.c may play a role in the growth and development of cucumber fruit and therefore become one of the crucial gene loci during breeding selection.

The DUF966 genes were found to be involved in the response of plants to environmental stress conditions. For example, the overexpression of *OsDSR2* (*DUF966-stress repressive gene 2*) in rice can dramatically increase the sensitivity of plants to salt and drought stress and reduce sensitivity to abscisic acid (ABA) [8]. In *Arabidopsis*, the DUF966 gene *AtST39* was found to improve salt and drought tolerance by positively regulating resistance pathways [8]. Consistently, in the present study, we found that CsDUF966_1.b, CsDUF966_5.b, and CsDUF966_6.a were regulated by the abiotic stress of 6 °C chilling, as well as the biotic stresses of downy mildew and nematode infection. Meanwhile, CsDUF966_4.c was responsive to low and high temperatures, short and long photo periods, and gibberellic acid, suggesting it plays a role in modulating the stress response of cucumber to adverse conditions. qRT-PCR analysis confirmed the response of DUF966 genes to different stresses, but the functions of these genes need to be further explored using molecular biology experiments.

The miRNAs are key post-transcriptional regulators [18]. They are involved in a broad range of plant growth, development, and stress response processes through cleavage and/or inhibition of translation of functional gene transcripts [19]. In this study, the post-transcriptional regulatory relationships of miRNAs targeting CsDUF966 transcripts were explored. As shown in Figure 6, ten cucumber miRNAs were predicted to target five CsDUF966 genes (CsDUF966_1.b, CsDUF966_4.c, CsDUF966_6.a, CsDUF966_7.c, and CsDUF966_Un3.c). All miRNAs regulate the expression of CsDUF966 through cleavage. Notably, CsDUF966_4.c was potentially targeted by eight miRNAs, including csa-miR-166, csa-miR-166k, csa-miR-166t-1, csa-miR-166t-2, csa-miR-166v, csa-miR-166u, csa-miR-166x, and csa-miR-164c.

The miR-164 plays a key role in flower and leaf development, lateral root initiation, seed expansion, and stress response [20]; for example, it was found to participate in regulating seed expansion in maize [20]. In cotton, miR-164 regulates lateral branching and plant architecture through the miR164-GhCUC2-GhBRC1 module [21]. In kiwifruits, miR-164 was found to regulate fruit ripening by targeting the transcription factors AdNAC6 and AdNAC7 [22]. In this study, we found that csa-miR-164c targeted CsDUF966_4.c, a gene proposed to contribute to the regulation of cucumber fruit development. Considering the known function of miR-164, it appears that csa-miR-164c and CsDUF966_4.c form a regulatory module that regulates cucumber fruit growth and development.

The miR166 is a highly conserved miRNA family that consists of multiple members in cucumbers [23,24]. In this study, we identified seven members targetingCsDUF966_4.c. The miR-166 plays critical roles in plant growth and development as well as in responses to biotic and abiotic stresses. It was reported that functions such as shoot apical meristem and floral development, and root growth in *Arabidopsis* [25,26,27]; fruit development, disease resistance, and adverse-temperature tolerance in tomato [28,29]; seed development, plant height, and cold, drought, and salinity stresses in soybean [23,30]; and salt stress in potato [31], were contributed to by miR-166. Considering the biological function of miRNA-166 members, it is believed that miRNA-166 and CsDUF966_4.c form a regulatory module that regulates cucumber growth and development, as well as stress response and tolerance.

CsDUF966_6.a is targeted by csa-miR-827c. miR827 binds to the 5′-UTR of *PHOSPHATE TRANSPORTER 5* (*PHT5*) homologs. Recent studies in both *Arabidopsis* and rice have demonstrated that PHT5 proteins function as vacuolar phosphorus (Pi) transporters that mediate Pi storage or remobilization [32]. In this study, we found that csa-miR-827c binds to the 3′-terminal of CsDUF966_6.a. Whether cucumber csa-miR-827c is involved in Pi uptake and storage by targeting CsDUF966_6.a still needs to be investigated. Our results reveal that CsDUF966_6.a is widely expressed in most cucumber cultivars, tissues, and treatments, and csa-miR-827c participates in the post-transcriptional regulation of CsDUF966_6.a expression, which lays the foundation for further function analysis of the csa-miR-827c-CsDUF966_6.a regulatory module.

## 4. Materials and Methods

### 4.1. Identification of CsDUF966 Family Members

The cucumber genome (ChineseLong_genome_v3) was downloaded from the CuGenDB (http://cucurbitgenomics.org/v2/organism/19/, accessed on 18 June 2022). The hidden Markov model (HMM) of the DUF966 domain (PF06136) was downloaded from Pfam (http://pfam.xfam.org/family/PF06136#tabview=tab3/, accessed on 18 June 2022) and used as query sequences to perform first-round BLASTp against cucumber reference protein sequences (e-value < 10^−10^) [33]. The reported DUF966 protein sequences were collected and used as query sequences to perform the second round of BLASTp [2]. After merging two rounds of BLASTp search results and deleting redundant sequences, unique sequences were further validated using Pfam (http://pfam.xfam.org/, accessed on 18 June 2022) and InterProScan (http://www.ebi.ac.uk/InterProScan/, accessed on 18 June 2022) to determine whether sequences contain the PF06136 domain and exclude sequences that do not contain the DUF966 domain [34].

### 4.2. Phylogenetic, Chromosomal Distribution, and Ka/Ks Analysis

Multiple sequence alignments (MSA) of CsDUF966 protein sequences were performed using ClustalW2 through the neighbor-joining (NJ) method with 1000 replicated-bootstraps [35]. A midpoint-rooted base phylogenetic tree was further illustrated using the online tool IToL (Interactive Tree of Life, http://itol.embl.de/, accessed on 18 June 2022) to infer the phylogenetic relationships of CsDUF966 and assist their nomenclature [36]. The corresponding gene annotation information for CsDUF966 was extracted from a GFF3 file. MapInspect was used to map CsDUF966 onto each chromosome according to the start and end positions [37]. The tandem and fragment duplicates were identified using the following rules. Tandem duplicate event fits the following criteria: (1) aligned sequence length > 80% regions of each sequence; (2) identity > 80%; (3) threshold ≤ 10^−10^; (4) only one duplication can be admitted when genes are linked closely; and (5) intergenic distance is less than 25 kb. Segmental duplication event fits the criteria of (1), (2), and (3) and is located on different chromosomes [38]. The Ka/Ks ratio (non-synonymous substitution rate/synonymous substitution rate) of CsDUF966 duplication pairs was calculated using TBtools [39]. For the evolutionary analysis of DUF966 genes among Cucurbitaceae species, 33 additional reference genomes belonging to cucumber, watermelon, melon, cucurbita, bottle gourd, sponge gourd, wax gourd, chayote, qingpiguo, and snake gourd were downloaded from CuGenDB (http://cucurbitgenomics.org/v2/, accessed on 18 June 2022) and used to identify DUF966 and calculate the Ka/Ks ratio following the methods mentioned previously.

### 4.3. Protein Characteristics, Gene Structure, and Domain Analysis

The basic characteristics of the UF966 proteins, including amino acid length, grand average of hydropathicity (GRAVY), molecular weight (MW), instability index, and isoelectric point (pI), were surveyed using the ExPASy Server10 (https://prosite.expasy.org/PS50011/, accessed on 18 June 2022). The signal peptide was predicted with SignalP5.0 (http://www.cbs.dtu.dk/services/SignalP/, accessed on 18 June 2022), and the subcellular localization was predicted using Plant-mPLoc (http://www.csbio.sjtu.edu.cn/bioinf/plant-multi/?tdsourcetag=s_pcqq_aiomsg/, accessed on 18 June 2022) [40]. Three-dimensional homology modelling was performed using the SWISS-MODEL online server (https://www.swissmodel.expasy.org/) [41]. According to the GFF3 annotation, CsDUF966 exon/intron structures were illustrated using TBtools [39]. Conserved motifs were identified through MEME (http://meme-suite.org/tools/meme/, accessed on 18 June 2022/, accessed on 18 June 2022) using the following parameters: each sequence contained any number of non-overlapping motifs, the maximum number of different motifs was 15, and the length of motifs ranged from six to 50 residues. The MEME output file was further illustrated using TBtools to visualize motif structures. The possible functions of the motifs were further surveyed using the Pfam online tool. The upstream chromosomal regions (1500 bp) were manually extracted from refence genome sequence and explored by PlantCARE (http://bioinformatics.psb.ugent.be/webtools/plantcare/html/, accessed on 18 June 2022) to identify the *cis*-elements in CsDUF966 promoters [42]. The number and type of *cis*-elements were sorted and displayed using the R package ‘pheatmap’ [43].

### 4.4. GO Annotation and Expression Pattern Profiling

Blast2GO was used to carry out the GO annotation [44]. The expression profiles of CsDUF966 were collected from the CuGenDB (http://cucurbitgenomics.org/ftp/RNASeq/cucumber/ChineseLong_v3/, accessed on 18 June 2022) and normalized as FPKM (fragments per kilobase of transcript per million fragments mapped values). Treatments were classified into growth and development stages, abiotic stresses, and biotic stresses. Then R package ‘pheatmap’ was used to produce the heatmap [43].

### 4.5. Prediction the miRNA Targeting to CsDUF966

To identify the potential miRNAs targeting to CsDUF966 transcripts, the cucumber mature miRNA sequences reported by Burkhardt and Day [24] were collected. Then, miRNA and CsDUF966 CDS sequences were submitted to the online tool psRNATarget with default parameters (https://www.zhaolab.org/psRNATarget/, accessed on 18 June 2022) [45,46]. The targeting relationships of miRNA to CsDUF966 were further visualized by R package ‘ggalluvial’ [47].

### 4.6. Subcellular Localization Analysis Assays

Following the method reported by Xiao, et al. [48], the subcellular localization of CsDUF966 proteins were experimentally explored. Briefly, the whole CDS regions of each coding gene were amplified by Phanta HS Master Mix (Vazyme, Nanjing, China). The amplicons were collected and connected to the plant expression vector pART27:GFP using a ClonExpress II One Step Cloning Kit (Vazyme). Positive clones were transformed into *Agrobacterium tumefaciens* GV3101 and transient expressed in the leaves of *Nicotiana benthamiana*. Sub-cellular localization was observed by fluorescence microscope (Olympus FV3000, Tokyo, Japan) after three days.

### 4.7. qRT-PCR Analysis

Cucumbers (*Cucumis sativus* L. cv. JinYou 1) were cultured in the chamber at 28 °C/18 °C day/night temperature using 1/2 strength Hoagland solution. At the two leaf stage, seedlings were subjected to five experimental groups: (i) control, seedlings were cultured using Hoagland solution; (ii) heat stress, seedlings were treated with 40 °C/32 °C day/night temperature in another identical growth chamber; (iii) chilling stress, seedlings were exposed to 18 °C/5 °C day/night temperature for chilling treatment; (iv) salt stress, 75 mM sodium chloride (NaCl); and (v) PEG stress, 20% PEG6000. The roots, stem, and leaf were collected after 3 days of treatment, respectively. The solution pH was maintained at 6.0 using 0.2 M HCl or 1 M KOH. All sampled materials were harvested to be frozen with liquid nitrogen and stored at −80 °C prior to RNA extraction.

Total RNA extraction from the samples and following cDNA synthesis were performed according to manufacturer’s instructions (Vazyme Biotech Co. Ltd., Nanjing, China). Real-time PCR analysis was performed using SYBR Green Master Mix (Vazyme Biotech Co., Ltd., Nanjing, China) on a CFX 96 Real-Time PCR system (Bio-Rad). The relative quantity was calculated with the 2^–ΔΔCt^ method. Primers were designed with Primer Premier 5.0 software (Appendix A) [49].

## 5. Conclusions

The CsDUF966 family members were systemically identified and characterized by the bioinformatics method. In total, eight CsDUF966 genes were identified from the cucumber genome. They were phylogenetically separated into three groups (a, b, and c). All CsDUF966 were hydrophilic proteins and were localized in nucleus. Both acidic and basic proteins were identified. An expression profiling analysis showed that CsDUF966 have variable expression patterns under ‘growth and development stage’, ‘abiotic stress’, and ‘biotic stress’ conditions. The CsDUF966_4.c and CsDUF966_7.c were under strong positive selection. Their expressions were precisely regulated in cucumber fruit and multiple miRNAs were players in the post-transcription regulation of CsDUF966_4.c and CsDUF966_7.c. Based on the neofunctionalization selection and expression patterns, CsDUF966_4.c and CsDUF966_7.c were speculated as fruit growth regulators and were strongly selected during the breeding program of cucumber.

## Figures and Tables

**Figure 1 plants-11-02497-f001:**
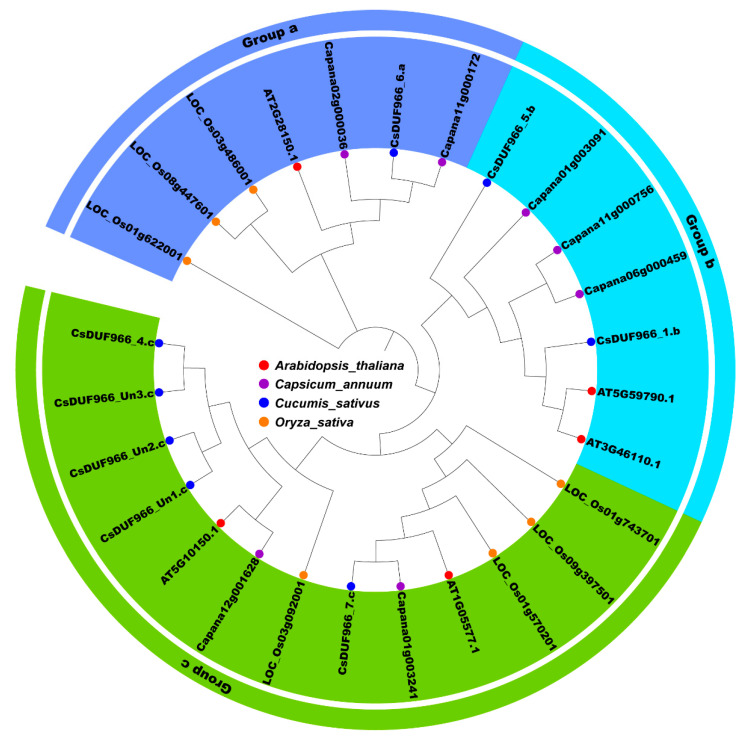
Phylogenetic analysis of DUF966 family members. The protein sequences were aligned using ClustalW2. Purple, group a; blue, group b; green, group c. Different colors of triangles represent DUF966 from *Arabidopsis thaliana*, *Capsicum annuum*, *Cucumis sativus*, and *Oryza sativa*.

**Figure 2 plants-11-02497-f002:**
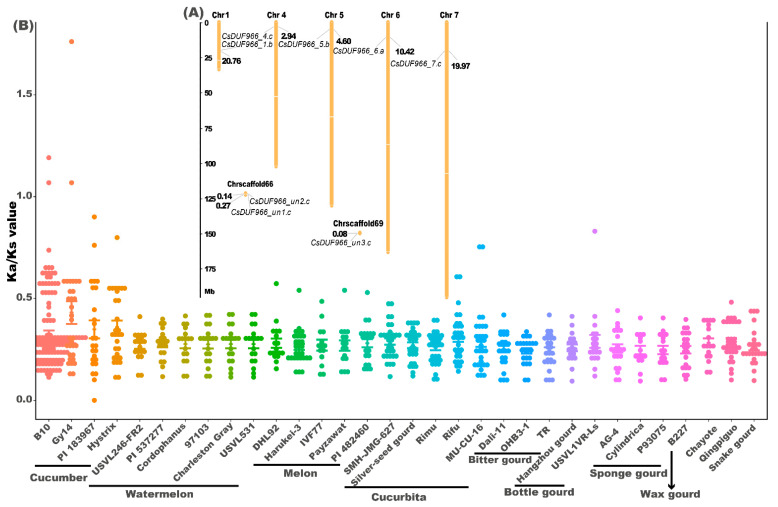
The (**A**) chromosome distribution of CsDUF966s and (**B**) Ka/Ks values of DUF966 homolog pairs between cucumber and other 33 Cucurbitaceae species/cultivars. The 34 genomes were download from CuGenDB (http://cucurbitgenomics.org/v2/, accessed on 18 June 2022).

**Figure 3 plants-11-02497-f003:**
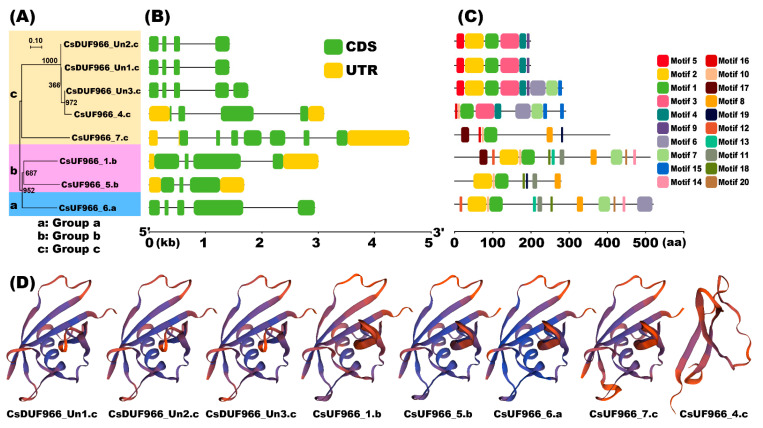
Characterization of CsDUF966. (**A**) Phylogenetic tree and groups. (**B**) Gene exon/intron structures. (**C**) Protein motifs. (**D**) Predicted 3D models of CsDUF966. Models were generated by using online website SWISS-MODEL.

**Figure 4 plants-11-02497-f004:**
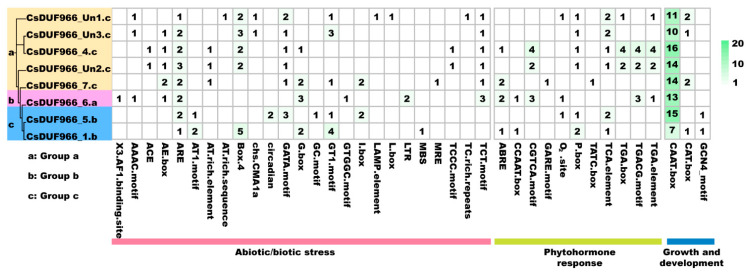
Analysis of *cis*-elements in promoter regions of CsDUF966 genes.

**Figure 5 plants-11-02497-f005:**
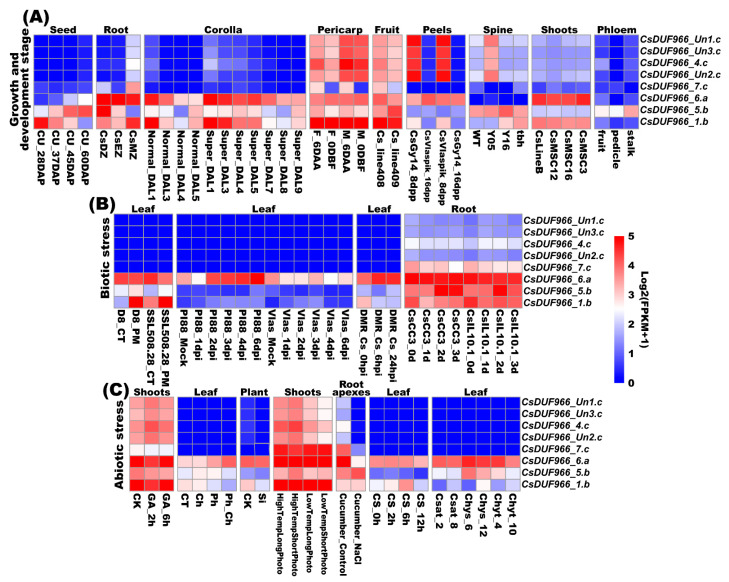
Expression pattern profiling of cucumber CsDUF966 genes. (**A**) CU_28DAP to 60DAP: cv. Marketmore seed at corresponding days after pollination. CsDZ, CsEZ, and CsMZ: cv. GY14 root differentiation, elongation, and meristematic zones at 4 days after plating. Normal_DAL1 to DAL5: corolla of normal ovaries at corresponding day after labeling of line 6457; Super_DAL1 to DAL9: corolla of super ovaries. F_6DBF and F_0DAA: cv. NCG122 pericarp tissue on the 6th day before flowering and ovary tissue on flowering day; M_6DBF and M_0DAA: cv. NCG121 tissues. Cs_line408 and Cs_line409: 2–4 cm early fruits from near-isogenic lines 408 (long fruit) and 409 (short fruit). CsGy14_8dpp and 16dpp: cv. Gy14 peels at 8- and 16-days post pollination; CsVlaspik_8dpp and 16dpp: cv. Vlaspik peels. WT and tbh: fruit spine of wild type and tiny branched hair mutant of cv. R1407; Y05 and Y16: spine from 0.5 and 1.6 cm long fruit. CsLineB, CsMSC3 to 16: shoots from cucumber line B, MSC3, MSC12, and MSC16 of 12-day-old plants. The fruit, pedicle, and stalk: cv. 9930 fruits, pedicle, and stalk phloem at 3 days post anthesis. (**B**) D8_PM and SSL508-28_PM: leaves of D8 and SSL508-28 lines inoculation with powdery mildew; D8_CT and SSL508-28_CT: control leaves. PI88_Mock to PI88_6dpi: cv. PI 197,088 mock and leaves infected with downy mildew at corresponding day post infection; Vlas_Mock to PI88_6dpi: cv. Vlaspik leaves. DMR_Cs_0hpi to 24 hpi: leaf tissues of downy mildew-resistant cv. PI 197,088 post inoculating with *Pseudoperonospora cubensis*. CsCC3_0d to 3d, and CsIL10-1_0d to 3d: root tissues of cv. CC3 and IL10-1 after nematode infection. (**C**) CK, GA_6h to 12h: cv. 13_3B shoot apices without treatment, and with GA treatment at corresponding hours. CT, Ch, Ph, and Ph_Ch: cv. Changchunmici control and leaves treated with chrysophanol, physcion, and both physcion and chrysophanol. Si and CK: line B10 in vitro plants in silicon addition and control medium. HighTempLongPhoto, HighTempShortPhoto, LowTempLongPhoto, and LowTempShortPhoto: cv. 9930 shoot tips under corresponding treatments. CS_0h to 12h: seedlings leaf tissues after 6 °C chilling. Chys_6 and 12, Chyt_4 and 10, and Csat_2 and 8: seedling leaf tissues of pickled cucumber, *Cucumis hytivus*, and cv. Beijingjietou. Detailed information can be found in CuGenDB (http://cucurbitgenomics.org/rnaseq/home, accessed on 18 June 2022).

**Figure 6 plants-11-02497-f006:**
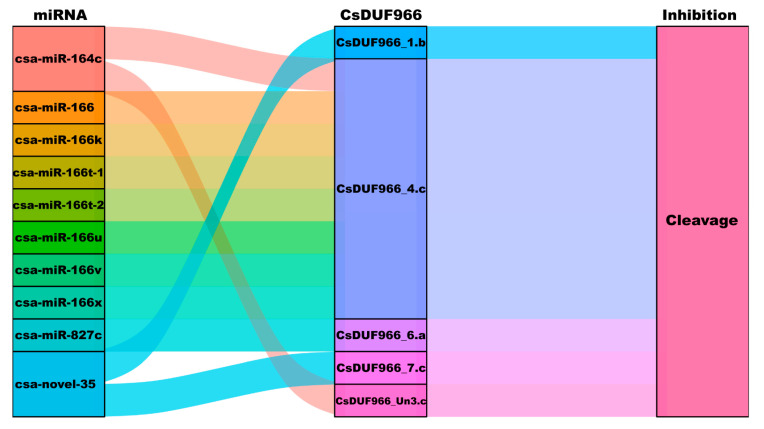
Sankey diagram for the relationships of miRNA targeting to CsDUF966 transcripts. The three columns represent miRNA, mRNA, and inhibition effect. Each rectangle represents one gene, and the connection degree of each relationship was visualized according to the size of the corresponding rectangle.

**Figure 7 plants-11-02497-f007:**
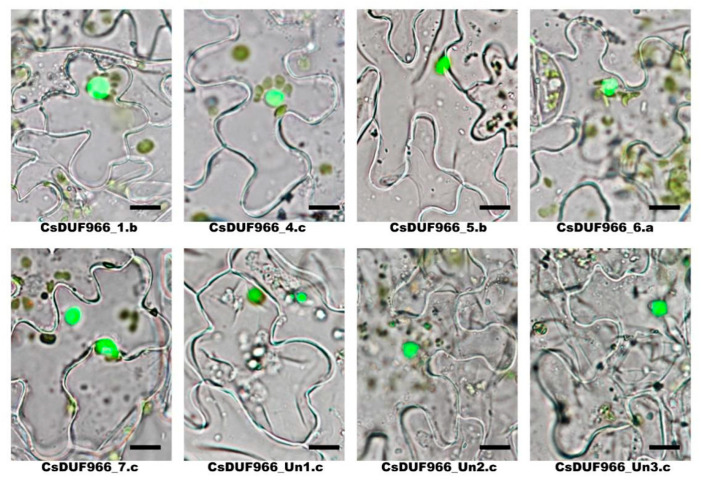
Subcellular localization of CsDUF966 proteins. Images were merged by bright field and green fluorescence (Bar = 25 μm).

**Figure 8 plants-11-02497-f008:**
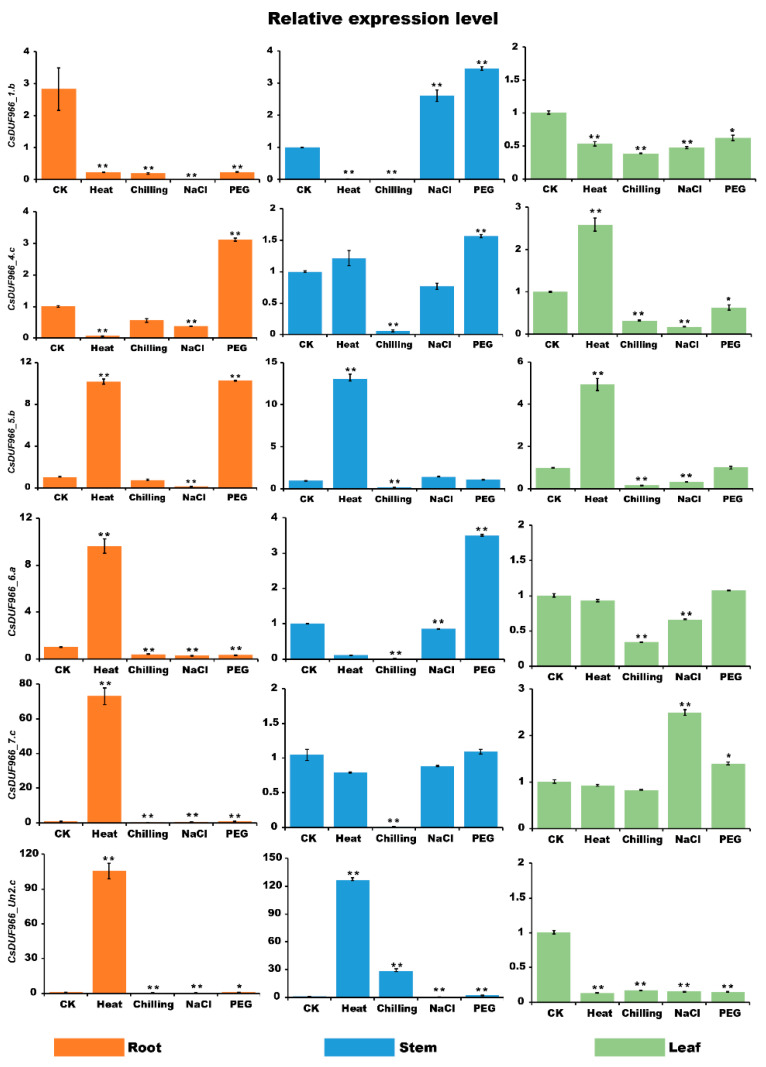
Relative expression profiles of CsDUF966 genes in cucumber after 3 days of treatment. Each bar represents the average of three replicates, and the error bar represents standard deviation (SD). Asterisk indicates significant differences at the level of 0.05.

**Table 1 plants-11-02497-t001:** Detail information of CsDUF966.

Name	Gene ID	Sub. Loc.	Len.	MW	pI	Stab.	Ali.	GRAVY
CsDUF966_1.b	CsaV3_1G033680.1	Nucleus	512	57.0	8.78	60.62	78.48	−0.554
CsDUF966_4.c	CsaV3_4G004680.1	Nucleus	291	33.0	5.31	55.48	56.56	−1.102
CsDUF966_5.b	CsaV3_5G006900.1	Nucleus	279	31.4	8.21	59.77	66.67	−0.78
CsDUF966_6.a	CsaV3_6G014330.1	Nucleus	520	58.0	7.95	64.91	62.62	−0.803
CsDUF966_7.c	CsaV3_7G031560.1	Nucleus	406	45.1	8.3	43.11	60.74	−0.865
CsDUF966_Un1.c	CsaV3_UNG146140.1	Nucleus	199	23.3	6.3	60.35	71.01	−1.048
CsDUF966_Un2.c	CsaV3_UNG137030.1	Nucleus	199	23.3	6.3	60.35	71.01	−1.048
CsDUF966_Un3.c	CsaV3_UNG218500.1	Nucleus	284	33.0	7.74	57.42	65.49	−1.035

Note: Sub. Loc.: subcellular localization; Len.: length of amino acid; MW: molecular weight (kDa); pI: isoelectric point; Ali.: Aliphatic index; GRAVY: grand average of hydropathicity.

## Data Availability

All datasets supporting the conclusions of this article are included within the article (and its additional files). The genome data and sequences and expression profiles of CsDUF966 genes used in the current study are available in the CuGenDB (http://cucurbitgenomics.org/ftp/RNASeq/cucumber/ChineseLong_v3/, accessed on 18 June 2022). The datasets generated and analyzed during the current study are available from the corresponding author on reasonable request.

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
