# Peer review of "Mining the Roles of Cucumber DUF966 Genes in Fruit Development and Stress Response"

_plants, 2022, doi:10.3390/plants11192497_

Round 1

Reviewer 1 Report

Dear Authors,

The MS Mining the roles of cucumber DUF966 genes in regulating fruit development and stress response” is original research topic which can be accepted for publishing after minor revision.

 Please, see some comments bellow

 Introduction part is missed real text of introduction about plant breeding problem under environmental stress effects. It would be good to say that during years were developed different methods of plant breeding - from methods of visual assessment to the new genomic technoloigies (Sytar et al., 2014, Villagómez-Aranda et al., 2022 ). New genomic technologies promise to make progress for breeding tolerance to the specific environmental stress (Witcomble et al., 2008).

The molecular genetics approaches like mutation, marker assisted selection (MAS), genomics, recombinant DNA technology, targeted induced local lesions in genome (TILLING), and virus induced gene silencing (VIGS) were adapted by breeders to develop effective resistance in crop plants in a shorter time (Haroon et al., 2022). 

 Please, see next references bellow

Sytar, O., Kosyan, A., Taran, N. et al. Anthocyanin’s as marker for selection of buckwheat plants with high rutin content. Gesunde Pflanzen 66, 165–169 (2014). https://doi.org/10.1007/s10343-014-0331-z

Witcombe JR, Hollington PA, Howarth CJ, Reader S, Steele KA. Breeding for abiotic stresses for sustainable agriculture. Philos Trans R Soc Lond B Biol Sci. 2008 Feb 27;363(1492):703-16. doi: 10.1098/rstb.2007.2179.

Villagómez-Aranda, A.L., Feregrino-Pérez, A.A., García-Ortega, L.F. et al. Activating stress memory: eustressors as potential tools for plant breeding. Plant Cell Rep 41, 1481–1498 (2022). https://doi.org/10.1007/s00299-022-02858-x

Haroon M, Wang X, Afzal R, Zafar MM, Idrees F, Batool M, Khan AS, Imran M. Novel Plant Breeding Techniques Shake Hands with Cereals to Increase Production. Plants. 2022; 11(8):1052. https://doi.org/10.3390/plants11081052

After L70 of Introduction would be good to add more detailed text line regarding aim of presented research with novelty. L392-393 Please, detiles which bioinformatics method.

Author Response

Dear Reviewer,

Many thanks for forwarding the comments and giving us an opportunity to improve our manuscript. All the comments are valuable and very helpful for us to improve our manuscript. The manuscript has been revised thoroughly according dear reviewers’ insight comments and valuable suggestions, and the revised areas were marked by yellow background in the text and deletion areas were indicated by tracing change.

Express my gratitude again and thank you for your help in improving this manuscript.

Best wishes to you and have a nice day.

Sincerely

Jun Liang Yin

---------------------------------------------------------------------------------------

Responds to the Reviewer 1:

The MS “Mining the roles of cucumber DUF966 genes in regulating fruit development and stress response” is original research topic which can be accepted for publishing after minor revision.

Please, see some comments bellow

Introduction part is missed real text of introduction about plant breeding problem under environmental stress effects. It would be good to say that during years were developed different methods of plant breeding - from methods of visual assessment to the new genomic technoloigies (Sytar et al., 2014, Villagómez-Aranda et al., 2022 ). New genomic technologies promise to make progress for breeding tolerance to the specific environmental stress (Witcomble et al., 2008).

The molecular genetics approaches like mutation, marker assisted selection (MAS), genomics, recombinant DNA technology, targeted induced local lesions in genome (TILLING), and virus induced gene silencing (VIGS) were adapted by breeders to develop effective resistance in crop plants in a shorter time (Haroon et al., 2022).

Please, see next references bellow

Sytar, O., Kosyan, A., Taran, N. et al. Anthocyanin’s as marker for selection of buckwheat plants with high rutin content. Gesunde Pflanzen 66, 165–169 (2014). https://doi.org/10.1007/s10343-014-0331-z

Witcombe JR, Hollington PA, Howarth CJ, Reader S, Steele KA. Breeding for abiotic stresses for sustainable agriculture. Philos Trans R Soc Lond B Biol Sci. 2008 Feb 27;363(1492):703-16. doi: 10.1098/rstb.2007.2179.

Villagómez-Aranda, A.L., Feregrino-Pérez, A.A., García-Ortega, L.F. et al. Activating stress memory: eustressors as potential tools for plant breeding. Plant Cell Rep 41, 1481–1498 (2022). https://doi.org/10.1007/s00299-022-02858-x

Haroon M, Wang X, Afzal R, Zafar MM, Idrees F, Batool M, Khan AS, Imran M. Novel Plant Breeding Techniques Shake Hands with Cereals to Increase Production. Plants. 2022; 11(8):1052. https://doi.org/10.3390/plants11081052

Response: Thank you very much for your valuable suggestion. More description has been added in Introduction as ‘Breeding is an effective way to improve crop yield and resistance to environmental stresses (Sytar et al., 2014, Villagómez-Aranda et al., 2022). The molecular genetics approaches like mutation, marker assisted selection (MAS), genomics, recombinant DNA technology, targeted induced local lesions in genome (TILLING), and virus induced gene silencing (VIGS) were adapted by breeders to develop effective resistance in crop plants in a shorter time (Haroon et al., 2022). In the past few years, the complete genome sequencing of cucumber and discovery of related genetic information have provided the opportunity to make progress for breeding tolerance to the specific environmental stress (Witcomble et al., 2008), and identify protein families that function in response to various stresses at the genome-wide level’. (L61-L69)

After L70 of Introduction would be good to add more detailed text line regarding aim of presented research with novelty. L392-393 Please, detiles which bioinformatics method.

Response: Thank you very much for your valuable suggestion. More text was added in Introduction as “Our results: (1) provide valuable information for understanding the classification and characterization of DUF966 genes, (2) reveal the roles of CsDUF966s in regulating cucumber fruit development and stress response, (3) and lay the foundation for further functional studies of CsDUF966s and their application in cucumber breeding programs.” (L74-78)

Reviewer 2 Report

Genome-wide analyses of cucumber DUF966 genes were performed to clarify the gene function in fruit development and stress response. Eight genes were identified in cucumber genome. However, nucleotide sequences of CsDUF966_Un1.c are almost identical to CsDUF966_Un2.c and CsDUF966_Un3.c, except the length of CsDUF966_Un3.c is longer than the other two. Authors should explain the relationship among these three genes in both nucleotide and amino acid sequences and the meaning of “Un”.

According to Fig. 5, expression of several cucumber DUF966 genes was induced by biotic or abiotic stress. No enough evidence to say these genes regulate fruit development and stress response in this study. RNA interference or mutants should applied to confirm their gene function.

Fig. 3(A) should be describe in the text. In Fig. 3(A), are “Group a~c” the same as the classification in Fig. 2? If yes, why cluster I is not Group c as indicated in the gene name? Why CsDUF966_Un1.c is not close to CsDUF966_Un2.c as in Fig. 2?

In Fig. 3(B), please explain the reason for some genes without UTR. It is easy to predict the end of 3’-UTR followed by polyadenylation signal. Also explain how to identify the length of 5’-UTR? Has the transcription start site been identified?

In Fig. 3(D), the 3D models of CsDUF966_Un1.c~ CsDUF966_Un3.c should not  be the same because CsDUF966_Un3.c is longer than CsDUF966_Un1.c and CsDUF966_Un2.c.

In Fig. 4, please explain the meaning of the values in the boxes. If the value represents the number of the cis-acting elements in the promoters, a gene has more than 25 TATA boxes is not reasonable. A promoter contains only one TATA box!

case-miR-164c were predicted to target CsDUF966_Un3.c. Will the mirNA also target to CsDUF966_Un1.c and CsDUF966_Un2.c since these three genes have almost the same sequences? Does cas-miR-164c target to the longer 3’-end of CsDUF966_Un3.c?

In Fig. 4, gene expression profiles were collected from the CuGenDB. It is better to be confirmed by qRT-PCR.

Author Response

Dear Reviewer,

Many thanks for forwarding the comments and giving us an opportunity to improve our manuscript. All the comments are valuable and very helpful for us to improve our manuscript. The manuscript has been revised thoroughly according dear reviewers’ insight comments and valuable suggestions, and the revised areas were marked by yellow background in the text and deletion areas were indicated by tracing change.

Express my gratitude again and thank you for your help in improving this manuscript.

Best wishes to you and have a nice day.

Sincerely

Jun Liang Yin

----------------------------------------------------------------------------------------

Responds to the Reviewer 2

Genome-wide analyses of cucumber DUF966 genes were performed to clarify the gene function in fruit development and stress response. Eight genes were identified in cucumber genome. However, nucleotide sequences of CsDUF966_Un1.c are almost identical to CsDUF966_Un2.c and CsDUF966_Un3.c, except the length of CsDUF966_Un3.c is longer than the other two. Authors should explain the relationship among these three genes in both nucleotide and amino acid sequences and the meaning of “Un”.

Response: Thank you very much for your valuable suggestion. As shown in Figure 2A, CsDUF966_Un1.c and CsDUF966_Un2.c are located in Chrscaffold66, and CsDUF966_Un3.c is located in Chrscaffold69. Un means genes in those contigs un-anchoring into chromosome. CsDUF966_Un1.c and CsDUF966_Un2.c have same nucleotide sequences, thus encode same amino acid sequences. Further considering the chromosome location and the distance between them (same scaffold, 130 Kb), CsDUF966_Un1.c and CsDUF966_Un2.c would be produced by gene duplication, which resulted in the gene copy variation. CsDUF966_Un3.c has same 5’-terminal with CsDUF966_Un1.c and CsDUF966_Un2.c, whereas the 3’-terminal of CsDUF966_Un3.c is longer than CsDUF966_Un1.c and CsDUF966_Un2.c, thus CsDUF966_Un3.c encodes longer protein. Further considering the chromosome location (different scaffold) and sequence similarity (same 5’- terminal and longer 3’- terminal), CsDUF966_Un3.c would be produced by gene duplication and stop codon mutant resulted in the longer 3’-terminal.

According to Fig. 5, expression of several cucumber DUF966 genes was induced by biotic or abiotic stress. No enough evidence to say these genes regulate fruit development and stress response in this study. RNA interference or mutants should applied to confirm their gene function.

Response: Thank you very much for your valuable suggestion. It true that the solid experimental evidence is not that enough to draw the conclusion that these genes regulate fruit development and stress response in this study. But to same extent, expression patterns imply their function. In most cases, genes expression pattern (temporal and spatial) and their function are closely related, in another word, the expression of certain genes in certain tissues suggested their function in these tissues. Consistently, in this study, the expression of CsDUF966_7.c was precisely regulated in fruits. It was highly expressed in peels 8 d post-pollination in cv. Gy14 and Vlaspik, but not in peels at 16 d. It was highly expressed in the pericarp tissue of cv. NCG121 and cv. NCG122 pericarp tissue on the 6th day before flowering, expressed at a low level in ovary tissue on the day of flowering, highly expressed in 2-4 cm early fruits of the long fruit near-isogenic line 408, and expressed at a low level in short fruit line 409 (Figure 5), suggesting that CsDUF966_7.c may participate in the regulation of cucumber fruit development. Furthermore, The CsDUF966_4.c and CsDUF966_7.c were under strong positive selection. Their expression were precisely regulated in cucumber fruit and multiple miRNAs were players in the post-transcription regulation of CsDUF966_4.c and CsDUF966_7.c. Based on the neofunctionalization selection and expression patterns, CsDUF966_4.c and CsDUF966_7.c were speculated as fruit growth regulators and were strongly selected during the breeding program of cucumber. (L290-298)

Furthermore, To explore the response of CsDUF966s to different stresses, the expression of CsDUF966s genes (CsDUF9661.b, CsDUF9664.c, CsDUF966_5.b, CsDUF966_6.a, CsDUF966_7.c, CsDUF966_Un2.c) under heat, chilling, salt stress and drought stresses were detected by qRT-PCR. And qRT-PCR analysis confirmed the response of DUF966 genes to different stresses. (L225-245)

Last but not least, based on the results in the present study, we selected several genes to performed their function analysis, but the results are still need to be perfected. If possible, we would like to put these results in the next article to make sure its integrity.

Fig. 3(A) should be describe in the text. In Fig. 3(A), are “Group a~c” the same as the classification in Fig. 2? If yes, why cluster I is not Group c as indicated in the gene name? Why CsDUF966_Un1.c is not close to CsDUF966_Un2.c as in Fig. 2?

Response: Thank you very much. We wrongly marked the group name by mistake. As far as “CsDUF966_Un1.c is not close to CsDUF966_Un2.c”, we repeated the procedure for phylogenetic tree, and realized the problem. We produced the tree file by ClustalW2, and visualized by IToL. As indicated in the following figure, CsDUF966_Un1.c is close to CsDUF966_Un2.c when using parameter “use branch length”. However, the tree is not well displayed. Thus we used the parameter “ignore branch length”, and thus it seems CsDUF966_Un1.c is not close to CsDUF966_Un2.c. To solve this problem, we used MEGA7 to visualize the phylogenetic tree, which obviously indicated CsDUF966_Un1.c is close to CsDUF966_Un2.c, and bootstrap values and tree scale were both displayed. And the wrongly marked group name was also corrected.

Use branch length

Ignore branch length

In Fig. 3(B), please explain the reason for some genes without UTR. It is easy to predict the end of 3’-UTR followed by polyadenylation signal. Also explain how to identify the length of 5’-UTR? Has the transcription start site been identified?

Response: Thank you very much for your insightful comment. The gene structure information including exon, intron, and UTR locations and lengths were extracted from the GFF3 file, which was downloaded from the CuGenDB, http://cucurbitgenomics.org/v2/organism/19. The website does not provide how did they identify the length of 5’-UTR, thus it is really difficult for us to explain.

However, after analysis of the GFF3 file, the reason for some genes with UTR and some without UTR are because the start codon of some gene is located in the middle position of the first exon, and the stop codon is located in the middle of last exon. In other word, those genes without 3’-UTR and 5’-UTR are because the start codon is located in the first position of the first exon, and the stop codon is located in the last position of last exon. And the full regions of 3’-UTR and 5’-UTR of each gene are not provided in the GFF3 file.

In Fig. 3(D), the 3D models of CsDUF966_Un1.c~ CsDUF966_Un3.c should not be the same because CsDUF966_Un3.c is longer than CsDUF966_Un1.c and CsDUF966_Un2.c.

CsDUF966_Un3.c is longer than CsDUF966_Un1.c and CsDUF966_Un2.c, it has longer C-terminal. Thus the real 3D structure of CsDUF966_Un3.c should be different from CsDUF966_Un1.c and CsDUF966_Un2.c.

However, the construct of 3D models of genes generally uses the same homology templet. Since the high sequence similarity among CsDUF966_Un3.c, CsDUF966_Un1.c and CsDUF966_Un2.c, it is possible that 3D models of these three CsDUF966 be the same.

In Fig. 4, please explain the meaning of the values in the boxes. If the value represents the number of the cis-acting elements in the promoters, a gene has more than 25 TATA boxes is not reasonable. A promoter contains only one TATA box!

Thank you very much for your reminding.

The number of different cis-acting elements were marked in the box.

It is true that a promoter contains only one TATA box. However, in this study and many other genes analysis study (Yi Qin and Wen Di et al., 2020; Genome‐wide analysis of ethylene‐insensitive3 (EIN3/EIL) in Triticum aestivum. Crop Sci. 21(561); Gao and Yang et al., 2020; Genome-Wide Identification of Metal Tolerance Protein Genes in Populus trichocarpa and Their Roles in Response to Various Heavy Metal Stresses." Int J Mol Sci 21(5).), the upstream chromosomal regions (1-1,500 bp) were manually extracted from reference genome sequence and explored by PlantCARE (http://bioinformatics.psb.ugent.be/webtools/plantcare/html/) to identify the cis-elements in CsDUF966 promoters, and many TATA box was detected. The reason for these results may due to: (1) the prediction is based on the similarity of query sequence to TATA box; thus some results are false positive; (2) 1.5 kb upstream chromosomal regions were used in many studies, which maybe longer than promoter region, thus resulted in the identification of multiple TATA box. Last but not least, the precise promoter region needs to be confirmed experimentally.

case-miR-164c were predicted to target CsDUF966_Un3.c. Will the mirNA also target to CsDUF966_Un1.c and CsDUF966_Un2.c since these three genes have almost the same sequences? Does cas-miR-164c target to the longer 3’-end of CsDUF966_Un3.c?

Response: Thank you for your reminding. The cas-miR-164c targets to the longer 3’-end of CsDUF966_Un3.c, thus cas-miR-164c does not targets to CsDUF966_Un1.c and CsDUF966_Un2.c.

In Fig. 4, gene expression profiles were collected from the CuGenDB. It is better to be confirmed by qRT-PCR.

Response: Thank you very much for your kind suggestion. To explore the response of CsDUF966s to different stresses, the expression of CsDUF966s genes (CsDUF9661.b, CsDUF9664.c, CsDUF966_5.b, CsDUF966_6.a, CsDUF966_7.c, CsDUF966_Un2.c) under heat, chilling, salt stress and drought stresses were detected by qRT-PCR. And qRT-PCR analysis confirmed the response of DUF966 genes to different stresses. (L225-245)

Reviewer 3 Report

Dear Authors congratulations for the work done that adds new informations to the topic, however paragraphs 2.5 and 2.6 are difficult to understand for the continuous reference to the acronyms,

I would suggest to review the aforementioned paragraphs avoiding the continuous recourse to the acronyms that generates confusion.

Best regards

Author Response

Dear Reviewer,

Many thanks for forwarding the comments and giving us an opportunity to improve our manuscript. All the comments are valuable and very helpful for us to improve our manuscript. The manuscript has been revised thoroughly according dear reviewers’ insight comments and valuable suggestions, and the revised areas were marked by yellow background in the text and deletion areas were indicated by tracing change.

Express my gratitude again and thank you for your help in improving this manuscript.

Best wishes to you and have a nice day.

Sincerely

Jun Liang Yin

-------------------------------------------------------------------------------------------------------

Responds to the Reviewer 3

Dear Authors congratulations for the work done that adds new informations to the topic, however paragraphs 2.5 and 2.6 are difficult to understand for the continuous reference to the acronyms,

I would suggest to review the aforementioned paragraphs avoiding the continuous recourse to the acronyms that generates confusion.

Best regards

Response: Thank you for your comment, acronyms has been checked and replaced in these section.

Reviewer 4 Report

In this manuscript, the authors characterized the eight members of cucumber DUF966. Most of the results were acquired by in silico analysis. In my view, this paper should be submitted to other suitable journals.  

I am wondering whether DUF966 members are involved in fruit development and stress response. I know public expression databases are important tools but, to validate the hypothesis, at least the authors should perform qPCR analysis. 

In the sections of abstract and conclusion,  the sentence " Both acidic and basic proteins were identified." is ambiguous.

In Figure 4, the authors should show the statistical significance in the numbers of cis-elements and should explain why TATA box is included in the category of "Growth and development".

Author Response

Dear Reviewer,

Many thanks for forwarding the comments and giving us an opportunity to improve our manuscript. All the comments are valuable and very helpful for us to improve our manuscript. The manuscript has been revised thoroughly according dear reviewers’ insight comments and valuable suggestions, and the revised areas were marked by yellow background in the text and deletion areas were indicated by tracing change.

Express my gratitude again and thank you for your help in improving this manuscript.

Best wishes to you and have a nice day.

Sincerely

Jun Liang Yin

-------------------------------------------------------------------------------------------------------

Responds to the Reviewer 4

In this manuscript, the authors characterized the eight members of cucumber DUF966. Most of the results were acquired by in silico analysis. In my view, this paper should be submitted to other suitable journals.  I am wondering whether DUF966 members are involved in fruit development and stress response. I know public expression databases are important tools but, to validate the hypothesis, at least the authors should perform qPCR analysis. In the sections of abstract and conclusion,  the sentence " Both acidic and basic proteins were identified." is ambiguous.

Response: Thank you for your comment. (1) To explore the response of CsDUF966s to different stresses, the expression of CsDUF966s genes (CsDUF9661.b, CsDUF9664.c, CsDUF966_5.b, CsDUF966_6.a, CsDUF966_7.c, CsDUF966_Un2.c) under heat, chilling, salt stress and drought stresses were detected by qRT-PCR. And qRT-PCR analysis confirmed the response of DUF966 genes to different stresses. The discritption was added as “As shown in the Figure 8, in the root, the expression levels of CsDUF966_5.a, CsDUF966_7.c and CsDUF966_Un2.c were greatly up-regulated under heat stress, but were decreased by chilling, cold, and PEG treatments. The expression levels of CsDUF966_1.b was decreased by all treatments. PEG treatments increased the expression levels of CsDUF966_4.c and CsDUF966_5.b. In the stem, the expression levels of CsDUF966_1.b, CsDUF966_4.c, CsDUF966_5.b, CsDUF966_6.a, and CsDUF966_7.c were largely decreased by cold stress. Heat stress decreased the expression levels of CsDUF966_1.b, CsDUF966_6.a, but increased the expression levels of CsDUF966_5.b and CsDUF966_Un2.c. PEG treatments increased the expression levels of CsDUF966_1.b, CsDUF966_4.c, CsDUF966_6.a. NaCl stress increased the expression levels of CsDUF966_1.b, but decreased the expression levels of CsDUF966_6.a and CsDUF966_Un2.c. In the leaf, the expression levels of CsDUF966_1.b and CsDUF966_2.c was decreased by all treatments. The expression levels of CsDUF966_5.b and CsDUF966_6.a were decreased by cold and NaCl treatments. Heat stress increased the expression levels of CsDUF966_4.c, CsDUF966_5.b, and greatly decreased that of CsDUF966_1.b and CsDUF966_2.c.” (L230-L245)

(2) This sentence has been revised as ‘Moreover, three acidic and five basic proteins were identified.’. (L18-19)

In Figure 4, the authors should show the statistical significance in the numbers of cis-elements and should explain why TATA box is included in the category of "Growth and development".

Response: Thank you for your comment. The number of different cis-acting elements were marked in the box. Considering the fact that the numbers of cis-elements without replication, it seems the statistical significance does not make sense.

In this study and many other genes analysis study (Yi Qin and Wen Di et al., 2020; Genome‐wide analysis of ethylene‐insensitive3 (EIN3/EIL) in Triticum aestivum. Crop Sci. 21(561); Gao and Yang et al., 2020; Genome-Wide Identification of Metal Tolerance Protein Genes in Populus trichocarpa and Their Roles in Response to Various Heavy Metal Stresses." Int J Mol Sci 21(5).), the TATA box is included in the category of "Growth and development". Accordingly, here we also following the category methods.

Round 2

Reviewer 2 Report

It is true that the solid experimental evidence is not enough to draw the conclusion that these genes regulate fruit development and stress response in this study” (quoted from the author's response). The expression of CsDUF966_7.c was regulated in fruits but it doesn't mean that the gene regulates fruit development. The gene is involved during fruit development. It’s better to revise the title of manuscript into “Mining the roles of cucumber DUF966 genes in fruit development and stress response”.

In author’s reply, for Fig. 3, “To solve this problem, we used MEGA7 to visualize the phylogenetic tree, which obviously indicated CsDUF966_Un1.c is close to CsDUF966_Un2.c, and bootstrap values and tree scale were both displayed”. But I saw the same phylogenetic tree as that in the previous version.

No matter how long upstream chromosomal regions were used, more than one TATA box indicated in Fig. 4 is nonsense. Of course prediction of PlantCARE is based on the similarity of query sequence to TATA box, therefore many putative TATA boxes will be detected easily. A scientist should think about it’s reasonable or not. Withdraw TATA box from Fig 4 shall be better.

Author Response

It is true that the solid experimental evidence is not enough to draw the conclusion that these genes regulate fruit development and stress response in this study” (quoted from the author's response). The expression of CsDUF966_7.c was regulated in fruits but it doesn't mean that the gene regulates fruit development. The gene is involved during fruit development. It’s better to revise the title of manuscript into “Mining the roles of cucumber DUF966 genes in fruit development and stress response”.

Response: Thank you very much for insight comment. We revised the title to “Mining the roles of cucumber DUF966 genes in fruit development and stress response” according to the valuable suggestion of dear Reviewer. (L2-3)

In author’s reply, for Fig. 3, “To solve this problem, we used MEGA7 to visualize the phylogenetic tree, which obviously indicated CsDUF966_Un1.c is close to CsDUF966_Un2.c, and bootstrap values and tree scale were both displayed”. But I saw the same phylogenetic tree as that in the previous version.

Response: We truly sorry for the mistake. It seems that, when replaces the Fig 3, we removed the inserted right one. The Fig 3 is replaced by the right one now. Thank you. (L139).

No matter how long upstream chromosomal regions were used, more than one TATA box indicated in Fig. 4 is nonsense. Of course prediction of PlantCARE is based on the similarity of query sequence to TATA box, therefore many putative TATA boxes will be detected easily. A scientist should think about it’s reasonable or not. Withdraw TATA box from Fig 4 shall be better.

Response: Thank you very much for your valuable suggestion. We removed the TATA box from Fig 4.

Reviewer 4 Report

Thank you for sending the revised manuscript. In future, I hope that the authors understand my point regarding the statistical analysis of cis-elements. I attached the file that I made to explain it. 

Author Response

Thank you for sending the revised manuscript. In future, I hope that the authors understand my point regarding the statistical analysis of cis-elements. I attached the file that I made to explain it.

Response: Thank you very much. We really appreciate the dear Reviewer for explaining the method for statistical analysis of cis-elements.